# *MCU* encodes the pore conducting mitochondrial calcium currents

**Dipayan Chaudhuri[1,2], Yasemin Sancak[3,4], Vamsi K Mootha[3,4], David E Clapham[2,5]\***

[1]Cardiovascular Research Center, Massachusetts General Hospital, Boston, United States; [2]Department of Cardiology, Howard Hughes Medical Institute, Boston Children's Hospital, Boston, United States; [3]Departments of Molecular Biology and Medicine, Massachusetts General Hospital, Boston, United States; [4]Department of Systems Biology, Harvard Medical School, Boston, United States; [5]Department of Neurobiology, Harvard Medical School, Boston, United States

**Abstract** Mitochondrial calcium ($Ca^{2+}$) import is a well-described phenomenon regulating cell survival and ATP production. Of multiple pathways allowing such entry, the mitochondrial $Ca^{2+}$ uniporter is a highly $Ca^{2+}$-selective channel complex encoded by several recently-discovered genes. However, the identity of the pore-forming subunit remains to be established, since knockdown of all the candidate uniporter genes inhibit $Ca^{2+}$ uptake in imaging assays, and reconstitution experiments have been equivocal. To definitively identify the channel, we use whole-mitoplast voltage-clamping, the technique that originally established the uniporter as a $Ca^{2+}$ channel. We show that RNAi-mediated knockdown of the *mitochondrial calcium uniporter* (*MCU*) gene reduces mitochondrial $Ca^{2+}$ current ($I_{MiCa}$), whereas overexpression increases it. Additionally, a classic feature of $I_{MiCa}$, its sensitivity to ruthenium red inhibition, can be abolished by a point mutation in the putative pore domain without altering current magnitude. These analyses establish that *MCU* encodes the pore-forming subunit of the uniporter channel.

**\*For correspondence:** dclapham@enders.tch.harvard.edu

**Competing interests:** The authors declare that no competing interests exist.

**Reviewing editor**: Richard Aldrich, The University of Texas at Austin, United States

## Introduction

Since the initial demonstration that mitochondria take up substantial amounts of cytoplasmic $Ca^{2+}$ (*Deluca and Engstrom, 1961*), detailed studies have revealed that this uptake can sculpt the cytoplasmic $Ca^{2+}$ transient (*Wheeler et al., 2012*), enhance ATP synthesis (*Balaban, 2009*), and trigger cell death (*Zoratti and Szabo, 1995*). Of several pathways for $Ca^{2+}$ entry, a uniporter found in the inner membrane possesses the largest capacity for uptake and was shown to be a highly $Ca^{2+}$-selective ion channel (*Kirichok et al., 2004*). However, despite this substantial progress, the identities of the genes encoding the functional uniporter were largely unknown until only recently. In the past several years, investigators from several laboratories have identified *mitochondrial calcium uptake 1* (*MICU1*, formerly *Cbara1*) (*Perocchi et al., 2010*), mitochondrial calcium uniporter (*MCU*, formerly *Ccdc109a*) (*Baughman et al., 2011*; *De Stefani et al., 2011*), and mitochondrial calcium uniporter regulator 1 (*MCUR1*, formerly *Ccdc90a*) (*Mallilankaraman et al., 2012*) as potentially integral components of this channel.

These discoveries have spurred an explosion of research into mitochondrial $Ca^{2+}$ transport, yet an outstanding question in this field is which, if any, of these several genes forms the pore subunit of the channel. So far, the discovery of each of these genes occurred via $Ca^{2+}$ imaging assays following RNAi-mediated knockdown. Although extremely useful for screening, this technique is ill-suited for separating modulatory effects from altered expression of the channel-forming subunit itself, and each of these genes was found to inhibit $Ca^{2+}$ uptake via this assay. This difficulty arises because $Ca^{2+}$ imaging measures $Ca^{2+}$ entry, which depends critically on (i) the number of open portals, (ii) the voltage

**eLife digest** Mitochondria are tiny organelles, less than a micrometre across, found inside almost all eukaryotic cells. Their main function is to act as the 'power plant' of the cell, generating adenosine triphosphate or ATP, which is the source of chemical energy for cellular processes. Beyond generating ATP, mitochondria perform many other functions: they contribute to various signalling pathways; they influence cellular differentiation; and they are involved in processes related to cell death.

Mitochondria are quite distinctive in appearance—they are enclosed by two membranes, a porous outer one and a largely impermeable inner membrane. Most mitochondrial functions involve proteins that control the movement of various molecules and ions across the inner membrane. One particularly important ion that must pass through this membrane is calcium; once inside the mitochondria, these calcium ions regulate cell survival and the generation of ATP.

Although several calcium import mechanisms exist, the best-studied pathway involves a pore-forming protein complex called the mitochondrial calcium uniporter. This ion channel has an exquisite selectivity, allowing only calcium into mitochondria even when other ions outnumber it a million-fold. Previously, researchers had identified several genes that are required for the formation of the uniporter, but it had not been established which of these encodes the central pore through which the calcium ions pass. Now, Chaudhuri et al. have shown that one of these—a gene called *mitochondrial calcium uniporter* (*MCU*)—codes for the protein subunit that creates the pore.

Prior studies used optical methods or purified proteins to study genes encoding the uniporter complex, producing controversial results regarding pore identity. This study uses a much more direct assay, namely electrophysiology performed on mitochondrial inner membranes. To access the inner membrane, the authors stripped off the outer membrane from whole mitochondria, and made them expand. By using a technique called voltage-clamping, Chaudhuri et al. were able to precisely measure calcium ion movement through intact or mutated channels. This technique controls confounding factors and minimizes the effect of contaminants that can plague interpretation of data acquired by other methods. They showed that blocking the expression of the *MCU* gene reduced the flow of calcium ions through the uniporter, whereas increasing *MCU* expression increased calcium transport.

One unique feature of the mitochondrial calcium uniporter is that it can be blocked by a dye called ruthenium red. Chaudhuri et al. used this property to confirm that the *MCU* gene encodes the pore-forming subunit of the channel complex—they identified a single point mutation in *MCU* that did not affect the channel's ability to transport calcium ions, but did abolish its sensitivity to ruthenium red. Together, these results show that the *MCU* gene encodes the pore of the mitochondrial calcium uniporter, and should lead to further research into the physiology and structure of this channel.

gradient, and (iii) the $Ca^{2+}$ concentration gradient driving flux, and none of these is controlled during imaging experiments. In fact, changes in $Ca^{2+}$ buffers or efflux pathways can alter matrix $Ca^{2+}$, and changes in pH or other divalents can alter indicator fluorescence, without reflecting a true difference in uniporter activity. Rather, the ideal method for assaying electrogenic ion transport is voltage-clamping, which allows precise control of the voltage and concentration gradient and thus accurate measurement of the changes in channel density.

Initial attempts to apply voltage-clamping focused on *MCU*, which possesses two transmembrane domains and highly-conserved acidic residues in a putative pore domain (*Baughman et al., 2011*; *De Stefani et al., 2011*; *Bick et al., 2012*). However, although MCU peptides reconstituted into lipid bilayers showed some activity in single-channel recordings (*De Stefani et al., 2011*), these experiments failed to recapitulate the typical single-channel behavior of the uniporter, most importantly its characteristic conductance and long-lasting openings (99% open probability) (*Kirichok et al., 2004*). Although such a discrepancy may be explained by the incorporation of the channel into bilayers in the absence of regulatory subunits, lipid bilayer reconstitution is also notorious for false positive findings. Since it detects

the activity of single molecules, channel-like behavior can be measured from trace contaminants after even crystallography-grade purification (*Accardi et al., 2004*). Thus, the identity of the pore-forming subunit remains to be definitively established.

To overcome the limitations of single-channel recordings, we chose to voltage-clamp whole mitoplasts (mitochondria stripped of their outer membrane and expanded osmotically to allow access to the inner membrane) (*Figure 1A*). This allows electrophysiological interrogation of all the channels assembled in their native milieu in individual mitochondria. Moreover, this technique allows us to express mutated channels and examine mitochondrial Ca$^{2+}$ currents ($I_{MiCa}$) for altered key features. Using this method, we show here that *MCU* does recapitulate key features of $I_{MiCa}$, and show that a single point-mutation can generate resistance to pharmacologic inhibition.

## Results

Stable transfection of HEK-293T cells with a short-hairpin RNA targeting *MCU* produced a substantial reduction in the protein when assayed by Western blot (*Figure 1B*) or quantitative real-time polymerase chain reaction (17 ± 5% transcripts remaining compared to shGFP). We isolated mitoplasts from these cells using the Kirichok protocol (*Fedorenko et al., 2012*; *Fieni et al., 2012*; *Figure 1A*). As expected from Ca$^{2+}$-imaging experiments (*Baughman et al., 2011*), mitoplasts from control cells showed robust $I_{MiCa}$ during voltage ramps from −160 mV to +80 mV (*Figure 1C*). Because $I_{MiCa}$ features a half-saturation value ($K_{0.5}$) of 20 mM [Ca$^{2+}$]$_{bath}$, we maximized current by recording in a 100 mM Ca$^{2+}$ gluconate bath solution (*Kirichok et al., 2004*). Utilizing high external Ca$^{2+}$ allows us to conclude that changes observed after modifying *MCU* expression is due to altered channel levels rather than modulation of $K_{0.5}$, which might be set by accessory subunits. Other critical features of $I_{MiCa}$ replicated in HEK-293T cells include its strong inward-rectification and high-affinity blockade by ruthenium red (RuR, 87 ± 2% inhibition in 100 nM RuR, *Figure 1C,E*).

Compared to the control condition, $I_{MiCa}$ in mitoplasts from shMCU-expressing cells was markedly smaller (*Figure 1D*). The RuR-sensitive component of total Ca$^{2+}$ current was reduced by 78 ± 14% (p<0.001, *Figure 1E*), with no significant difference in the RuR-insensitive residual component,

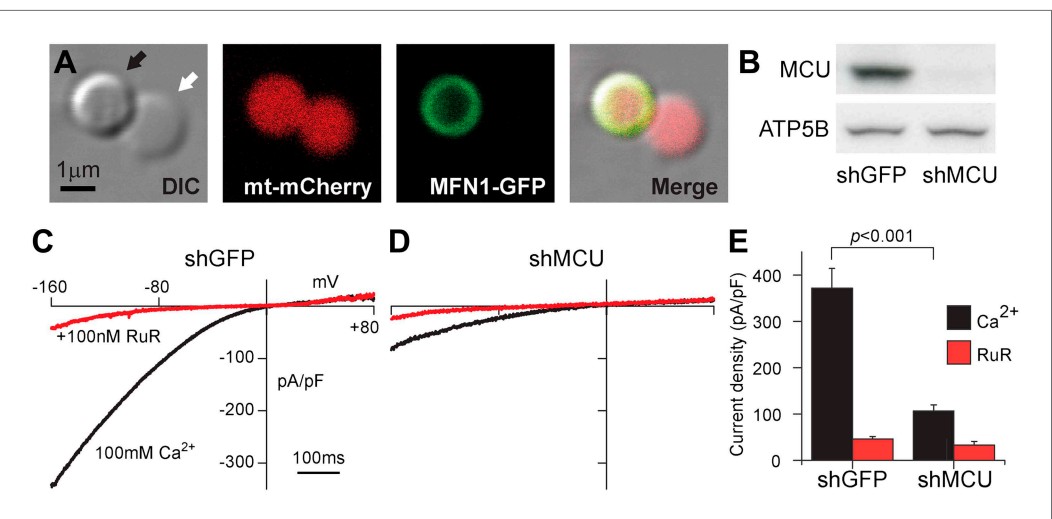

**Figure 1**. *MCU* expression recapitulates $I_{MiCa}$. (**A**) A HEK-293T cell mitoplast under differential interference contrast (far left), showing a typical figure-eight shape. The lobe bounded only by the mitochondrial inner membrane appears less dense (white arrow) than the lobe also bounded by the outer membrane (black arrow). Matrix-targeted mCherry demonstrates that the inner membrane forms a surface on both lobes (middle left). GFP-tagged mitofusin-1 (an outer membrane GTPase, middle right) reveals the outer membrane partially encapsulating only one lobe. Far right: merged image. (**B**) Western blots demonstrate reduced MCU expression following short-hairpin RNA-mediated knockdown of *MCU* but not *GFP* (control). ATP5B is a loading control. (**C**) Exemplar current traces demonstrate endogenous $I_{MiCa}$ in control knockdown (black trace), which is largely blocked by 100 nM RuR (red trace). Voltage ramps from −160 to +80 mV for 750 ms were delivered every 6 s from a holding potential of 0 mV. (**D**) Significantly reduced $I_{MiCa}$ after *MCU* knockdown. (**E**) Summary data, *n* = 6–10 per condition, error bars report SEM.

suggesting that the knockdown was specific to $I_{MiCa}$ and not a generalized reduction in membrane conductance. Moreover, differences were not due to alterations in mitochondrial structure, as mitoplast capacitance, a surrogate for inner membrane surface area (100µm²/pF), was consistent across all conditions tested here (shGFP: 0.48 ± 0.10 pF, shMCU: 0.34 ± 0.09 pF, p>0.05).

Next, we examined if overexpression of wild-type or mutant human MCU proteins substantially changed $I_{MiCa}$. We focused on an MCU mutated to encode an alanine at a highly-conserved serine in the putative pore-forming loop (S259A) (*Baughman et al., 2011*; *Bick et al., 2012*). The wild-type and mutant MCU proteins were tagged with a carboxyl-terminal FLAG epitope, and localized appropriately to mitochondria (*Figure 2A,B*). To confirm targeting to the inner membrane, we treated HEK-293T mitochondrial fractions with increasing concentrations of digitonin, and assayed for enzymatic digestion with proteinase K (*Baughman et al., 2011*; *Figure 2C,D*). Since the FLAG-tagged proteins were protected from digestion to a comparable extent as peptidyl-prolyl cis-trans isomerase F (PPIF, a matrix protein), whereas proteins of the outer- and inter-membrane spaces were not, the MCU constructs are appropriately targeted with their carboxyl-termini facing the mitochondrial matrix. Turning to our whole-mitoplast recordings (*Figure 2E,G*), we found that overexpression of wild-type MCU-Flag produced a robust increase in $I_{MiCa}$ of approximately 3.4-fold compared to endogenous HEK-293T currents (compare to *Figure 1C,E*). This enhanced $I_{MiCa}$ retained its sensitivity to RuR (*Figure 2E,G*).

Finally, we studied the S259A-MCU mutant to see if it disrupted key features of $I_{MiCa}$. In Ca²⁺-imaging experiments, this mutant had preserved Ca²⁺ uptake but diminished sensitivity to uniporter

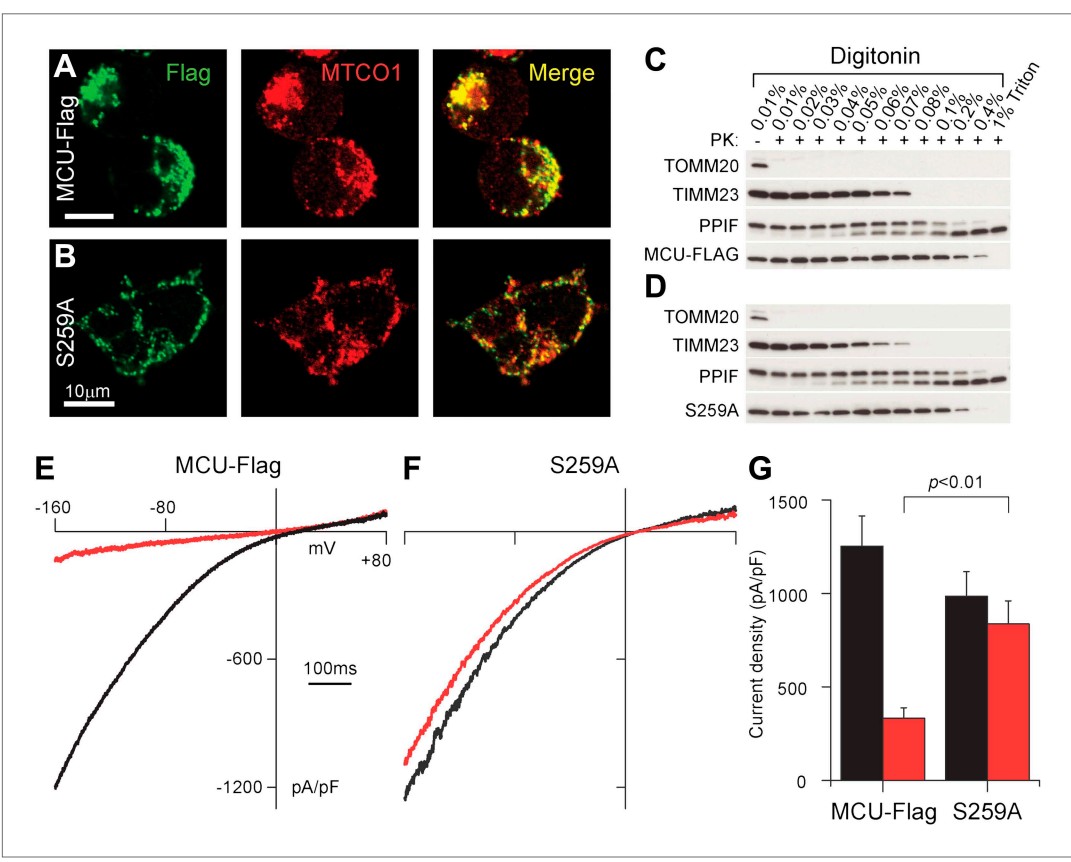

**Figure 2**. *MCU mutants alter $I_{MiCa}$ sensitivity to RuR*. (**A**) Confocal imaging of HEK-293T cells for FLAG-tagged MCU (left) and cytochrome C oxidase I (a mitochondrial marker, center). The merged image (right) shows colocalization. (**B**) As in (**A**) but for MCU-259A-FLAG. (**C**) Immunoblot analysis of proteinase K digestion after wild-type MCU-FLAG-expressing HEK-293T mitochondrial fractions are digitonin-permeabilized at the specified concentrations, confirming mitochondrial targeting. (**D**) Proteinase K digestion analysis for MCU-S259A-FLAG mutant. (**E**) Significant increase in $I_{MiCa}$ after MCU-FLAG overexpression. (**F**) Loss of RuR block in S259A mutant. (**G**) Summary data, n = 5–9 per condition.

inhibitors (*Baughman et al., 2011*). At baseline, mitoplasts from HEK-293T cells transfected with the S259A mutant had capacitances similar to mitoplasts after wild-type MCU overexpression (MCU-Flag: 0.32 ± 0.06 pF, S259A: 0.29 ± 0.10 pF, p>0.05). Moreover, S259A overexpression mirrored the increase in $I_{MiCa}$ seen after wild-type MCU transfection, confirming a fully-functional channel (*Figure 2F,G*). However, this variant displayed markedly decreased sensitivity to RuR, with minimal inhibition at 100 nM. Since overexpression occurred on a background of endogenous channels, this mutant appears to act in a dominant-negative fashion. In particular, the S295A RuR-inhibited fraction (148 ± 33 pA/pF, *Figure 2G*) was much less than the RuR-inhibited fraction in endogenous $I_{MiCa}$ (372 ± 42 pA/pF, *Figure 1F*). This suggests that, since the channel is an oligomer of multiple MCU subunits (*Baughman et al., 2011*), mutant S259A-MCU can incorporate with endogenous, wild-type subunits to form a $Ca^{2+}$-conducting channel, which nonetheless has drastically-reduced RuR sensitivity.

## Discussion

In our study, we use the most direct assay available to measure mitochondrial $Ca^{2+}$ currents—whole-mitoplast electrophysiology (*Kirichok et al., 2004*). By isolating mitoplasts and recording directly from all the channels embedded in the inner membrane, we measure the ensemble current and thus avoid confounding minor contaminants. This method, moreover, is the same technique that originally demonstrated that the mitochondrial $Ca^{2+}$ uniporter was an ion channel, allowing us to make a direct comparison to the characteristic features of this current ($I_{MiCa}$), and to determine if the genetically-modified channels could alter them.

We focused on MCU, the leading candidate for the pore-forming subunit, as it possesses two transmembrane domains and has highly-conserved acidic residues in a putative pore-like domain. First, we show that reducing *MCU* transcripts via knockdown, or enhancing them through overexpression, produce parallel changes in $I_{MiCa}$. Second, we show that a central feature of $I_{MiCa}$—its exquisite sensitivity to blockade with ruthenium red—can be abolished via a single point mutation in the pore domain. MCU subunits carrying this mutation act in a dominant-negative fashion, producing functional channels that conduct $Ca^{2+}$ ions but are largely insensitive to ruthenium red. We conclude that the functional uniporter pore at the mitochondrial inner membrane is formed by MCU multimers. We believe these experiments put to rest the outstanding question of the pore identity, firmly establishing that *MCU* produces $I_{MiCa}$.

## Materials and methods

### Cell culture

HEK-293T were obtained from ATCC and grown in Dulbecco's modified Eagle medium with high glucose supplemented with 10% fetal bovine serum and penicillin/streptomycin (Life Technologies, Grand Island, NY). HEK-293T cells (250,000 cells per well) were infected with lentivirus expressing short hairpin RNA (for knockdown experiments) or wild-type or mutant FLAG-tagged *MCU*. 2 days after infection, cells were split and selected with the appropriate antibiotic. Stable HEK-293T cells lines expressing shGFP and shMCU were maintained in 2 μg/ml puromycin (Sigma, St. Louis, MO), while MCU-FLAG and S259A lines were maintained in 200 μg/ml hygromycin B (Sigma).

### Plasmids

Vectors for expressing shRNA (pLKO.1) were obtained from the Broad Institute's RNAi Consortium (Sigma). To silence *MCU*, we used a hairpin targeting the 3′ UTR, with sequence 5′-GCAAGGAGTTTCTTT CTCTTT-3′ (TRCN0000133861). The control hairpin was against GFP, with sequence 5′-ACAACAGCC ACAACGTCTATA-3′ (shGFP) (TRCN0000072181). The FLAG-tagged full-length human *MCU* cDNA (NM_138357.1) was cloned into the pLJM5 vector (*Sancak et al., 2010*). The *MCU* mutant S259A was generated with QuikChange according to the manufacturer's instructions (Agilent, Santa Clara, CA). Human Mitofusin-1 (NM_033540) was cloned out of HEK-293T cell cDNA (see Quantitative polymerase chain reaction section, below) and ligated in-frame into the pEGFP-N3 vector (Clontech, Mountain View, CA) using KpnI and BamHI restriction sites. Matrix-targeted mCherry was engineered by replacing the GFP construct of the pAcGFP1-Mito plasmid (Clontech), using BamHI and NotI restriction sites.

### Quantitative polymerase chain reaction

Primers against human *MCU* were designed using PRIMER-Blast (NCBI). The forward primer was 5′-TTCCTGGGACATCATGGAGC-3′, and the reverse primer was 5′-TGTCTGTCTCTGGCTTCTGG-3′.

For qPCR, confluent HEK-293T cells were resuspended in TriZOL reagent (Life Technologies), and total RNA was subsequently purified following the manufacturer's instructions. 2 µg of RNA was used to synthesize cDNA using SuperScript ViLo (Life Technologies) and 100 ng of this was used per well for qPCR. Reactions were run in triplicate using SYBR Green (Agilent) on an Eppendorf Mastercycler epgradient S Realplex4 qRT-PCR cycler (Eppendorf, Hauppauge, NY). Analysis compared expression in shMCU cells to shGFP cells using the $2^{-\Delta\Delta Ct}$ method, normalizing to *glyceraldehyde 3-phosphate dehydrogenase* expression.

## Western blots

10 µg of protein from cell lysates or 5 µg of protein from crude mitochondrial preparations were loaded on 12% or 18% Tris-Glycine gels, proteins were transferred to PVDF membranes and then incubated with the indicated antibodies at 4°C overnight. We used antibodies against TOMM20 (sc-17764; Santa Cruz Biotechnology, Dallas, TX), TIMM23 (No. 611222; BD Biosciences, San Jose, CA), PPIF (ab110324; Abcam, Cambridge, MA), ATP5B (ab14730; Abcam), FLAG epitope (No. 2368; Cell Signaling Technology, Danvers, MA), and MCU (polyclonal chicken antibody raised against residues 41–351 of human MCU tagged with a 6His epitope; Covance, Dedham, MA).

## Proteinase K protection assay

Two confluent plates of HEK-293T cells were washed with PBS and the cells scraped into 1.5-ml microcentrifuge tubes in 1 ml of final volume. The cells were lysed by passaging through a 27.5 G needle several times. Samples were spun at 800$g$ for 10 min at 4°C to pellet nuclei and non-lysed cells. The supernatant was removed and spun separately at 8000$g$ for 10 min at 4°C to pellet the mitochondrial fraction. Mitochondria were resuspended in a sucrose buffer (200 mM sucrose, 10 mM Tris-MOPS, 1 mM EGTA-Tris) to a final concentration of 1 mg/ml. 20 µg of mitochondria were treated with digitonin (0.01–0.4%, Sigma) or 1% Triton-X100 (Sigma) and 100 µg/ml of proteinase K for 15 min at room temperature, in a 30 µl final volume. Proteinase K was then inhibited by addition of 5 mM PMSF. 5 µg of protein was loaded to Tris-glycine gels for Western blotting.

## Imaging

Stably-transfected HEK-293T cells were washed in PBS, which was used as the base solution in all following steps. Cell were fixed in 4% formaldehyde, permeabilized in 0.5% Triton-X100, and blocked in 10% goat serum, 0.1% Tween-20 (Biorad, Hercules, CA). Primary antibodies were a rabbit anti-FLAG and a mouse anti-MTCO1 (a mitochondrial marker, ab14705; Abcam). Secondary antibodies were Alexa-488 conjugated to a goat anti-rabbit, and Alexa-546 conjugated to goat anti-mouse (Life Technologies). Cells were imaged on an Olympus Fluoview 1000 confocal microscope (Center Valley, PA).

For mitoplast images, HEK-293T cells were co-transfected with mt-mCherry and MFN1-GFP plasmids using Lipofectamine 2000 reagent (Life Technologies) according to the manufacturer's instructions. Mitoplasts were isolated (as described in 'Whole-mitoplast recording') 2 days following transfection, and imaged on the confocal microscope as above.

Brightness levels have been slightly adjusted in the images to improve contrast. This adjustment has been applied to each image as a whole and has not obscured or eliminated any particular feature of the image. All reagents were from Sigma unless otherwise stated.

## Whole-mitoplast recording

We adapted prior methods to HEK-293T cells (*Fedorenko et al., 2012*; *Fieni et al., 2012*). Cells were grown to confluency on three 15-cm dishes. All subsequent manipulations were done using ice-cold solutions and equipment. Cells were scraped onto divalent-free PBS and spun down to pellet cells. The cell pellet was subsequently resuspended in a sucrose solution (250 mM sucrose, 5 mM HEPES, 1 mM EGTA, 0.1% BSA). The suspension was homogenized using an overhead stirrer. Centrifugation at 700$g$ pelleted nuclei. The supernatant was transferred to a new tube and centrifuged at 8500$g$ to pellet a crude mitochondrial fraction. This pellet was resuspended in a hypertonic mannitol solution (440 mM mannitol, 140 mM sucrose, 5 mM HEPES, 1 mM EGTA) for 10 min. The suspension was then passed through a French Pressure Cell homogenizer (Thermo Scientific, Waltham, MA) at 2000 p.s.i. to disrupt the mitochondrial outer membrane. The eluate was centrifuged at 10,500$g$ to pellet mitoplasts. Mitoplasts were stored in a hypertonic solution (750 mM KCl, 100 mM HEPES, 1 mM EGTA) on ice and protected from light.

For recordings, mitoplasts were aliquoted into the initial bath solution (150 mM KCl, 10 mM HEPES, 1 mM EGTA) to allow them to expand for 20 min. An individual mitoplast was visualized and attached to the pipette tip with gentle suction. Pipettes were formed from borosilicate glass (Sutter, Novato, CA) with tip resistances of 20–60 MOhm. We used sodium gluconate for pipette internal solutions (150 mM sodium gluconate, 10 mM HEPES, 2 mM EDTA, with sucrose to achieve an osmolarity of 410–430 mOsmol). Electrophysiology was performed using an Axopatch 200B amplifier (Molecular Devices, Sunnyvale, CA). Upon giga-ohm seal formation, pipette capacitance transients were cancelled, and the inner membrane was ruptured by 10–50 ms pulses (350–1000 mV). After establishing the whole-mitoplast configuration, a series of 4 ms pulses of 20 mV were recorded to allow curve fitting for capacitance measurements. Bath solutions were subsequently exchanged for a high-$Ca^{2+}$ solution (100 mM calcium gluconate, 20 mM HEPES, 2 mM $CaCl_2$, with sucrose to achieve osmolarity of 295–305 mOsms) and high-$Ca^{2+}$ +100 nM RuR solution (Sigma). RuR solutions were prepared fresh for every experiment. Solution exchanges were rapid, as prolonged matrix exposures to $Ca^{2+}$ caused activation of a significant leak component. Comparisons were made via the Student's *t*-test.

## Acknowledgements

We would like to thank Francesca Fieni, Andriy Fedorenko, and Yuriy Kirichok for assistance with the whole-mitoplast protocol, and Andrew Markhard for technical assistance.

## Additional information

### Funding

| Funder | Grant reference number | Author |
| --- | --- | --- |
| National Institutes of Health | F32HL107021 | Dipayan Chaudhuri |
| National Institutes of Health | R24DK080261 | Vamsi K Mootha |
| Helen Hay Whitney Foundation | | Yasemin Sancak |
| Howard Hughes Medical Institute | | David E Clapham |

The funders had no role in study design, data collection and interpretation, or the decision to submit the work for publication.

### Author contributions

DC, YS, Conception and design, Acquisition of data, Analysis and interpretation of data, Drafting or revising the article; VKM, DEC, Conception and design, Drafting or revising the article

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
