## [Decision Letter]

Thank you for sending your work entitled “*MCU* encodes the pore conducting mitochondrial calcium currents” for consideration at *eLife*. Your article has been favorably evaluated by a Senior editor and 3 reviewers, one of whom is a member of our Board of Reviewing Editors.

The following individuals responsible for the peer review of your submission want to reveal their identity: Richard Aldrich (Reviewing editor); Jon Lederer (peer reviewer).

The Reviewing editor and the other reviewers discussed their comments before we reached this decision, and the Reviewing editor has assembled the following comments to help you prepare a revised submission.

This work for the first time addresses structure-function relationships within the mitochondrial Ca^2+^ uniporter using the patch–clamp technique. Although indirect optical methods helped to identify three candidate proteins important for the activity of the mitochondrial Ca^2+^ uniporter (*MICU1*, *MCU*, and *MCUR1*), the subsequent attempts to understand the exact functions of these proteins led to little progress and produced controversial results. In this manuscript, the authors provide strong evidence that one of the candidate proteins, MCU, forms the pore of the mitochondrial Ca^2+^ uniporter. Since this is the first structure–function study of the *MCU* using the patch–clamp technique, methodologically this works sets an important precedent in the field.

Overall, the manuscript is well written and provides sufficient evidence to suggest that the MCU protein is a pore-forming subunit of the mitochondrial Ca^2+^ uniporter. There are two important points to be addressed:

1) The quality of the image of an 8-shaped mitoplast on Figure 1 is rather low. From this image it is not clear whether this is a real 8-shaped mitoplast or two attached swollen round-shaped mitoplasts. Also, in addition to the mitoplast(s), the DIC image contains debris that complicates our understanding. It would be useful to provide an image of higher quality.

2) Figure 1 and Figure 2: It is important to describe in the main text or in the figure legends how these traces were obtained. Are these responses to a voltage ramp protocol? What was the exact voltage protocol used? If a ramp protocol was used, it would perhaps be somewhat misleading to say that these traces are real I/V curves (as in the legend to Figure 1).

---

## [Author Response]

*1) The quality of the image of an 8-shaped mitoplast on Figure 1 is rather low. From this image it is not clear whether this is a real 8-shaped mitoplast or two attached swollen round-shaped mitoplasts. Also, in addition to the mitoplast(s), the DIC image contains debris that complicates our understanding. It would be useful to provide an image of higher quality*.

We have replaced the mitoplast image from Figure 1 with the new one shown below.

For this image, we co-transfected HEK cells with a plasmid encoding a mitochondrial matrix-targeted mCherry fluorescent protein, as well as a plasmid encoding GFP-tagged mitofusin-1, a mitochondrial outer membrane GTPase. This image is free of debris, and demonstrates that the structure is one mitoplast. We have replaced the figure legend as follows:

“A HEK-293T cell mitoplast under differential interference contrast (far left), showing a typical figure-eight shape. The lobe bounded only by the mitochondrial inner membrane appears less dense (white arrow) than the lobe also bounded by the outer membrane (black arrow). Matrix-targeted mCherry demonstrates that the inner membrane forms a surface on both lobes (middle left). GFP-tagged mitofusin-1 (an outer membrane GTPase, middle right) reveals the outer membrane partially encapsulating only one lobe. Far right: merged image.”

*2) Figure 1 and Figure 2: It is important to describe in the main text or in the figure legends how these traces were obtained. Are these responses to a voltage ramp protocol? What was the exact voltage protocol used? If a ramp protocol was used, it would perhaps be somewhat misleading to say that these traces are real I/V curves (as in the legend to Figure 1)*.

We have taken out the mention of I/V curves, and added the following to the figure legend to describe the voltage command protocol:

“Voltage ramps from −160 to +80 mV for 750 ms were delivered every 6 seconds from a holding potential of 0 mV.”